# The Task Specification Problem

**Pulkit Agrawal**
Improbable AI Lab, MIT
`pulkitag@mit.edu`

**Abstract:** Robots are commonly used for several industrial applications and some have made their mark even in households (e.g., the roomba). Undoubtedly these systems are impressive! However, they are very narrow in their functionality and we are faraway from building a robot butler. A central challenge is the ability to work with sensory observations and generalization to novel situations. While we do not prescribe a solution to this problem, we do provide a perspective on a few dominant ideas in robot learning for multi-task learning and generalization. This perspective suggests a counter-intuitive conclusion: the primary challenge in building generalizable robotic systems (e.g., a robot butler) is not in the learning algorithms or the hardware, but how humans transfer their knowledge into robots.

**Keywords:** Transfer Learning, Multi-task Learning, Commonsense

One dream is to build robots that can perform general household tasks (i.e., a robot butler). What is stopping us? There are several hypotheses: dexterous control is hard, state-of-the-art computer vision systems fail to infer object segmentation or 3D-pose accurately, deep networks are not robust, long-term planning is challenging, lack of suitable tactile sensors, continual and transfer learning do not work as well, safety in unstructured environments is a significant concern, reinforcement learning is data-hungry, etc. Often the robot's inability to perform household tasks is also attributed to the lack of *commonsense*. While *commonsense* broadly refers to a human-like understanding of the world, its definition remains subjective and eludes rigorous mathematical formalism. While investigations into various challenges listed above led to tremendous progress, we are far away from a household robot.

The central challenge in household environments is learning a diverse set of tasks and generalization. In other domains, such as language(e.g., GPT-3) and image processing [1], learning from large data has yielded impressive results. Similarly a viable approach to building a robot butler is by collecting large data-sets of demonstrations for diverse tasks that are then distilled into a behavior policy. However, the expense of collecting demonstrations is a major impediment. Other options are self-supervised data collection [2, 3, 4], providing alternative task descriptions such as rewards/goals and letting the robot collect data to maximize task performance. Whether data collection can be scaled is intimately tied to the generalization of the learned policy during data collection itself.

Our central insight is that *under-specification* – a situation where a problem admits multiple solutions underlies many core challenges not only in task-communication and policy generalization, but also plagues commonly used methods to overcome under-specification such as building in priors, regularization and feature learning. It simply re-appears in different forms. Under-specification manifests as reward-hacking in RL, the inability to infer intent from demonstrations, the learning of non-robust features that do not transfer. Intuitively incorporating prior knowledge should mitigate under-specification. However, counter-intuitively under-specification remains a formidable challenge in implementing well-known priors employed by humans. We will now elaborate our arguments.

## 1 Under-specification in Task Communication

Consider the problem of instructing the robot to fetch a bottle of water from the refrigerator. Suppose the robot is in the bedroom. An intuitive way to communicate the task is by saying, "Get a bottle of water." Completing this task requires the robot to understand the meaning of *get*, what is a bottle, the specific location of the bottle (e.g., in a refrigerator), how to get to the refrigerator, open it and take the bottle out. The point is that the language instruction *under-specifies* the task. Background knowledge is necessary for interpreting the instruction and planning action commands. One possibility is building in this knowledge that we are advocating as the central challenge in building generalizable robotic systems. A second option is instruct actions at every time step. But, it defeats the purpose of language!

Blue Sky Papers, 5th Conference on Robot Learning (CoRL 2021), London, UK.

Another way to communicate a task is by defining a reward function. Sparse rewards provide little guidance to the agent and are often insufficient for learning. On the other hand, dense rewards are subject to *reward hacking* [5], i.e., learning behaviors that optimize the reward function in an undesirable manner. A well-known example is of a cleaning robot [6]. Suppose it is rewarded for the amount of dust it picks up. Instead of learning to clean the entire room (the actual intent), the agent can maximize reward by simply repeating the process of throwing and picking up dust in one corner of the room. This behavior does not result from optimization failure but because multiple equally good solutions are admitted by the reward function (i.e., under-specification). Forcing the agent to learn the desired behavior requires changing the reward function, which is a tedious process. Unsurprisingly reward functions for popular benchmarks like meta-world [7], OpenAI-Gym are nothing short of meticulously crafted small programs. Outside standard benchmarks, the practice of learning a new task often devolves into an ad-hoc and laborious process of determining the reward function that results in the desired behavior with the algorithm at hand. Without additional priors, no algorithm can overcome this issue because it is a fundamental consequence of under-specification.

An alternative is to use sparse rewards but overcome the challenge in exploration using either a manually designed curriculum [8, 9] or a task ordering that is guided by intrinsic rewards [10, 11, 12, 13, 14], sub-goals [15, 16] generated by an agent or competition between agents [17]. Like reward engineering, manual curriculum design is tedious and largely dependent on human intuition about good task ordering. Intrinsic rewards such as prediction error [14], visit counts [13], learning progress [18], etc. are simply heuristics that encourage the agent to explore in a task-agnostic manner. In some cases, such exploration aligns with the task reward, and in others, it is not. It is therefore unsurprising that a recent study concluded that no intrinsic reward method consistently improves performance on ATARI games [19]. Further, on games where intrinsic rewards help, different reward formulations were suitable for different games. Similar criticism applies to the curriculum discovered by the competitive play between agents or by automatic goal generation. In summary, the success of exploration using these methods hinges on the *hope* that the self-generated curriculum will align with the desired task. Moreover, even when these methods work, meticulous tuning of parameters such as relative weighting of intrinsic v/s task rewards is required.

Instead of specifying the task with a reward function, one can provide the robot with a goal-state or a task demonstration. Goal-conditioned policies can be trained using supervised learning of inverse models [3, 20, 21, 22] or by reinforcement learning [23]. The goal can be a low-dimensional state vector describing the robot and the objects or an image, or even a video demonstration. One substantial challenge is in interpreting the goal itself. E.g., a goal image showing a set of stacked objects has multiple interpretations. The intended goal might be to stack objects in a particular order; it might be that the order is irrelevant. It is even possible that the arrangement of objects in a stack is irrelevant, and the actual goal was just to place objects on the table. A goal image is an incomplete description of the task. Prior knowledge about task distribution is required to infer the task correctly.

One might argue that the image interpretation problem can be simplified by observing a video demonstration of the entire task. For instance, if a demonstrator carefully stacks objects, the likely goal is not to put objects on the table but to stack them. However, even such an interpretation requires additional assumptions about the demonstrator, like completing the task in the shortest time or expending the least amount of energy or that he/she is rational. Such assumptions are often used to interpret human intent (e.g., pragmatics [24]). While these assumptions can rule out a few inferences, they still admit numerous goal interpretations even when used with a video demonstration. To gain intuition, consider a video demonstration depicting a hand pushing an object. If one reasons at the pixel level, it might be concluded that the goal is to move the arm instead of pushing the object. It is because the arm is larger than the object, and its motion is therefore dominant. From the same video, humans would interpret the task as pushing because of their implicit prior knowledge that the motion of objects is more significant than the arm itself. However, as we will discuss in Section 3, incorporating such priors presents another set of challenges.

The issue of intent inference from a single demonstration has been tackled using meta-learning on many demonstrations of different variants of the task [25] or using inverse models [20, 21]. These methods only work on a narrow set of videos (see meta-learning discussion in Section 2). Inverse reinforcement learning (IRL) infers intent by matching the distribution of states in the demonstration [26, 27]. Therefore its success inevitably relies on the feature representation of states.

In summary, under-specification affects different forms of task communication. One possible solution to accurately infer the intended target form goals specified as an image, language or a demonstration relies on learning good feature representations. We discuss this approach in the section below.

## 2    *Under-specification* in Representation and Transfer Learning

A distinctive characteristic of household environments is the diversity in tasks, the objects to be manipulated, the variation in object arrangements, and even their appearance! In such a setting, *transfer* is critical for success. In addition to aiding task communication, learning good representation is touted to overcome the transfer problem. Learning transferable representations by maximizing reward signals [28, 29] or even by predicting actions such as in learning from demonstrations has proven to be challenging. One alternative is to learn features by training on a proxy task such as image classification on Imagenet, a technique that has proven very successful for computer vision applications [30]. However, due to the differences in the proxy and desired task, the proxy features may not transfer. E.g., image classification is invariant to the pose or location of the object. However, these are precisely the features required for manipulation. Therefore when learning features on a proxy task, one is simply "hoping" that these features will transfer!

Even unsupervised and self-supervised methods that leverage large amounts of unlabelled data suffer from such "hopeful" generalization! For instance, learning disentangled features by reconstructing the input is a popular feature learning method. The underlying hope is that *disentangled features* will encode semantic image characteristics such as the lighting, shape, texture, etc. When trained on datasets with low visual complexity (e.g., MNIST), these methods successfully find semantic features. However, as the visual complexity increases, disentanglement is not unique. Many different sets of disentangled features result in the same reconstruction loss (i.e., under-specification). Some desirable, others not. E.g., JPEG compression learns a basis of Fourier features that perfectly reconstruct an image. However, these features are not as useful for manipulation or semantic understanding tasks.

In self-supervised learning, numerous proxy tasks have been proposed for feature learning [31, 32, 33, 34, 35]. It is worth noting that proxy tasks are inspired by human intuition and not from theoretical principles. The fact that the learned features generalize to popular computer vision tasks is merely an empirical finding! In other words, the task designer *simply hoped* that optimizing the proxy task learns transferable features. Suppose one encounters a task to which these features do not transfer. In that case, there is no guidance on what to do except for manually constructing new proxy tasks.

*Meta-learning* [36, 37] holds the promise of overcoming hopeful generalization by directly optimizing for representations that transfer. But such training requires apriori knowledge of the test-time task distribution. Because task-similarity has no formal definition and is subjective, constructing a parametric distribution of test tasks is impossible. Consequently, specifying the test-time distribution entails enumerating a list of all possible test tasks, which is infeasible for many practical applications.

The above discussion highlights core challenges in representation learning: lack of theory to guide the construction of proxy tasks and learning of unwanted features. The vital point to realize is that undesirable features are not a consequence of learning failure or because deep networks find a local minimum. Today's deep networks can perfectly optimize the training loss even for a random dataset [38]. Unwanted features are learned because many distinct solutions result in similar training losses. Among these, the desirable and undesirable solutions (dubbed as "non-robust" features [39]) are equally good in the absence of additional priors. The human inability to *precisely* specify what neural networks should learn (i.e., under-specification) results in training objectives that are too unconstrained. Consequently, they admit a large number of (potentially undesirable) solutions.

## 3    *Under-specification* in Incorporating Priors

Under-specification is a challenge because it leads to undesirable solutions. It is essential to realize that *there is no fundamental reason that makes a solution undesirable*. It is just that some solutions do not align with human intuition. E.g., consider a dataset with images of the digit two colored in red and six in blue. Both color and shape are equally good features for classification, yet we prefer shape over color. Because test distribution is unavailable during training, a desirable solution is not an objective but an arbitrary, subjective choice from a learner's perspective. Intuitively prior knowledge should mitigate such under-specification. However, the role of priors is not to rule out any subset of solutions but the specific ones that are not human-aligned. Because human alignment is subjective, even defining priors becomes a challenge. To be clear, we are not the first to suggest that priors are important, but to reveal that just any prior is not enough – it must be human aligned.

To understand the problem in designing such priors, consider regularization in machine learning. Widely used regularization penalties are inspired from the *occam's razor* of encouraging simpler solutions. However, these heuristics might also prune functions that generalize in ways similar to humans and be counter-productive. For instance, deeper neural networks with more parameters are better (and more human-aligned) at object recognition than DNNs with fewer parameters (i.e., simpler in the conventional sense). Another example is the prior of *diversity* for learning distinct skills [40] or encouraging *high-entropy* solutions. The problem is that even two random sequences of actions are nearly orthogonal in the action-space and therefore "diverse". What humans care about are "specific" diverse solutions. E.g., in walking, these would be forward, backward, left, or right motion, and not just any set of random action sequences are diverse. Diversity by itself is under-specified! Furthermore, human intuition about priors, especially with deep learning, is questionable. In addition to over-parameterized networks performing better, even commonly held architectural priors such as convolution have recently been brought to question with transformers [41].

How can we build human-like priors? One option is a knowledge base [42], which have proven unsuccessful for operation *in-the-wild*. Other possibility is to incorporate priors used by humans such as reasoning in 3D space instead of 2D images, representing the world in terms of objects, using models such as Newtonian physics that predict the future for planning etc. Despite their appeal it turns out these priors cannot be defined universally in a task-agnostic way. Consider the following:

**What is an Object?** : Consider the task of moving a jar full of candies. To solve this task, representing the entire jar as an object is logical. If the task is changed to count the number of candies, then each candy must be treated as an object. Further, if someone wants to eat the candy, then the candy wrapper and the candy should be grouped as two different objects. Because of this, computer vision researchers have also grappled with the definition of objects for a long time. The current consensus is to have humans mark object boundaries [43]. But, different people mark different boundaries! These boundaries are infact averaged to form the ground truth labels. While humans can flexibly group different pixels into an object depending on the task, the state-of-the-art object segmentation systems are overfit to one specific definition provided by the dataset. This is a problem, because there is no universal task-agnostic definition of an object, rather objects may only be defined in context of tasks.

**What is 3D/Geometry?** : When driving a a useful notion of 3D is the distance from vehicle in front is useful. For grasping objects surface normals provide critical information about grasp points. Whereas, the shape of objects is often defined in terms of a point cloud or a voxel grid. Again, just like objects, depending on the task, different interpretations of 3D are apt.

**What is a Model?** : When we think of models, we usually think of forward models that forecast future states. These models are apt for some tasks: e.g., physics is useful for pushing objects. However for other tasks, alternate formulations of a "model" are more apt. For navigation an occupancy map or spatial relationships between landmarks is a useful model. When searching for an object, a model indicating usual location of objects (e.g., pens in drawers) is useful. Similarly, different models are useful for playing chess (game rules) or video games (jump on platforms, kill enemies, etc.).

These examples clearly show that even popular priors inspired by humans reasoning are under-specified. They can only be made concrete in context of a task and their realization is task-dependent.

## 4   The Path Forward: New Approaches to Human Aligned Learning

We advocate that in addition to algorithms and data, how to transfer human knowledge is a fundamental challenge in robot-learning and resolving it will be key to realizing a robotic butler. One method for incorporating human knowledge that has worked well is *data augmentation* – where a human designer explicitly identifies what properties of the data are irrelevant. Future work should look at developing ideas that can directly transfer what is human relevant. In terms of teaching robots it might mean a paradigm shift where we don't leave the robot with a dataset in the hope that the learning algorithms will find the right solution. Instead, interacting with the learning systems akin to communication between mother and child over the lifetime of learning might be critical. At the level of deep learning architectures this may involve building shallower brain inspired architectures that have recurrence/feedback and leverage the *embodiment* prior. The point of this paper is not to suggest a solution, but to elaborate the problem of under-specification and how it manifests. The goal of building a robot butler suggests that we need machine learning that optimizes for human-alignment and not just the task performance. As a final comment its worth mentioning that there are also drawbacks of learning human-aligned learning: it becomes harder to learn superhuman solutions!

**Acknowledgments**

We thank members of the Improbable AI lab and the Embodied Intelligence group at MIT for many discussions that inspired this paper. The ideas in this work were developed while working on research sponsored by the DARPA Machine Common Sense Program, IBM, Sony, Amazon, Salesforce, Toyota Research Institute, US Airfoce's AI Accelerator at MIT, MIT's Systems That Learn Program, and the NSF Institute of Artificial Intelligence and Fundamental Interactions (IAIFI).

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
