# OpenReview forum: "The Task Specification Problem "
_robot-learning.org/CoRL/2021/Conference/Blue_Sky — CoRL 2021, Blue Sky_

### Official Review · Reviewer_2vpT · 2021-08-27

**Novelty:** Good
**Impact:** 4
**Clarity Of Presentation:** Excellent

**Recommendation:**

Weak Accept: I recommend accepting the paper, but will not argue for my recommendation if the majority of other reviewers have a different opinion.

**Summary:**

The paper discusses a fundamental problem in robot learning, the fact that most (if not all) current techniques for training robots under-specify the objectives of learning in a way that undermines generalization and limits the learned function (or control) to controlled and closed environments. The paper uses the example of building robots that can perform general household tasks to illustrate the problem of under-specification. To perform such tasks, the robot needs to learn policies (or reward functions) that can generalize beyond the provided demonstrations (or specified goal states). To do so, the robot needs to learn features that are pertinent to the desired task. However, current RL or imitation learning techniques fait to infer intents from demonstrations (or desired goal states). The paper argues that this problem should be solved through the incorporation of priors. For instance, if the demonstration shows a human hitting an object with their arm and pushing it forward, most people would understand that the intent is to push the object, not to move the arm in a specific way. This is due to the fact that humans know a prior that the movement of the arm is a waste of energy unless it accomplishes something in the environment.
The paper argues that this same problem occurs in all forms of data provided to robots to learn from it, such as reward functions, video demonstrations, or task communication. The paper also argues that underspecification is a key issue in representation and transfer learning. The paper concludes by emphasizing the necessity to incorporate human-aligned priors in robot learning.
The paper is well-written and enjoyable to read. The discussed problem is indeed critical, and we need new solutions to it in robot learning in order to realize the dream of building multi-purpose robots that can operate in everyday environments. I would have appreciated a little bit more elaborate discussion of the potential solutions to this problem and how to automatically learn "human-aligned" priors. The paper suggests that humans can explicitly provide these priors through communication, but this can be impractical if one wants to embody all human knowledge in such priors. A self-supervised approach seems more appropriate.
Finally, I would like to point out that the paper follows a rather informal style in writing. In particular, exclamation marks are overused, which is not appropriate for a technical text.

**Summary Of Recommendation:**

The paper is overall good and includes stimulating and thought-provoking discussions. The writing is however a little bit informal and could be made more rigorous. A more extensive list of potential solutions to the problem would have been appreciated.

---

### Official Review · Reviewer_bczu · 2021-08-28

**Novelty:** Fair
**Impact:** 3
**Clarity Of Presentation:** Very Good

**Recommendation:**

Weak Reject: I recommend rejecting the paper, but will not argue for my recommendation if the majority of other reviewers have a different opinion.

**Summary:**

This paper aims to argue that "the primary challenge in building generalizable robotic systems (e.g., a robot butler) is not in the learning algorithms or the hardware, but how humans transfer their knowledge into robots." Authors state that their "central insight is that under-specification – a situation where a problem admits multiple solutions underlies many core challenges in robot learning."

The paper begins by describing how it is challenging to define the task that we want a robot to learn. For example, reward functions are difficult to engineer, goal images only consider the final state of the environment, and demonstrations are often sub-optimal. The paper then moves away from robotics and discusses machine learning in general. Here, the focus is on how with transfer learning, it is difficult to specify the intention of pre-training or meta-learning, without over-constraining to the test data, and that often the best solutions are empirically motivated rather than theoretically motivated.

The paper then argues that we should be developing priors in machine learning that are more human-aligned, rather than using the standard priors such as "simplicity" or "diversity". The paper then notes that it is difficult to engineer these priors since defining certain human-interpretable concepts (e.g. "What is an object?") cannot easily be done in a general, task-agnostic way.

The paper then concludes with some high-level and somewhat vague ideas of how we should be proceeding with machine learning and robot learning.

**Summary Of Recommendation:**

I found the paper to be an interesting read, and it did make me think a little about the challenges in machine learning. However, I found that the paper has a lack of focus and, rather than building up to an interesting conclusion or an interesting set of questions, it kept jumping around between different ideas, without ever really building up any momentum towards a particular "blue-sky" idea. I was quite interested in the paper at the start, but by the end, it became clear that the paper mainly just summarises a number of challenges in machine learning, under the umbrella of "task specification". However, I found the use of "task specification" throughout the paper to bit of a pretence, and it seems to be used to just describe the difficulty in setting up a machine learning problem, rather than any particularly novel interpretation of what "task specification" means.

A few more specific comments:


1) The paper starts off nicely by talking about robot learning. However, the narrative quickly moves quite far away from robotics and just became a discussion on machine learning in general. It would have been good to have focussed more specifically on robotics, because this is where I think "task specification" is actually a good topic to consider. In robotics, it is genuinely difficult to communicate to a robot what the task is that it should learn. However, in machine learning, since we work in code and mathematics, this communication is actually very simple: we just write down the code or the mathematics. Yes, there can still be multiple solutions that the machine learning may find, but this is a very different kind of "task specification" to defining a task for a robot to solve in the real world. So by the time the paper moves away from robotics and machine learning, it is not very clear what "task specification" really means. It comes across as just being an umbrella term for all the design choices we have to make in machine learning, and so this part of the paper ends up being rather broad and vague.


2) As the authors say, "The point of this paper is not to suggest a solution, but to elaborate the problem of under-specification and how it manifests." But even though it was interesting reminding myself about all the problems in machine learning, the paper always felt like it was leading up to something which never materialised. The paper could be a good "Introduction" to a longer paper, but as a standalone paper it is quite "negative": there are lots of criticisms of machine learning, but no real solutions. I think that "blue sky" papers should be doing more than revisiting known issues and re-writing them under some pretence such as "task specification". The end of the paper does propose some solutions, but they are very vague: "Future work should look at developing ideas that can directly transfer what is human relevant", and "interacting with the learning systems akin to communication between mother and child over the lifetime of learning might be critical."


3) I do not agree that "diversity by itself is under-specified", or at least, I think this is missing the point. The point of diversity is that it prevents us from overfitting pre-training / exploration to a specific task / local minimum in a reward function. It encourages solutions that may otherwise not have been found if pre-training or exploration were too short-sighted. In the paper's walking example, what if we wanted the robot to walk up the stairs, and it only knew how to walk forwards, backwards, left, and right? Here, forcing human-like priors overly constrains the set of future solutions. The authors do mention this limitation of over-engineering priors, but this weakens the paper's motivation for advocating human-like priors. There were a few other examples in the paper where a problem with machine learning was proposed, then a solution was proposed, and then a limitation of that solution was proposed, which prevents the paper from building up any momentum.


4) It becomes quite confusing what "under-specification" really means. It is defined by the authors as "a situation where a problem admits multiple solutions", but there are so, so many causes for this. In the example with the images of the coloured digits, I don't consider this to be a problem of under-specification. I consider it to be a problem of not having training data which reflects the testing data. If a network is trained to predict the number in the image, then we are specifying the problem perfectly well, it's just that the data is not sufficient to cover all the possible test scenarios. Ok, so we could describe a dataset that is too small as "under-specification", but then almost all machine learning issues can be described as "under-specification". By then, the definition of "under-specification" has broadened out so far from the narrower interpretation I took from the "Get a bottle of water" example in the beginning of the paper.


5) Although much of the paper focusses on machine learning rather than robotics, I would have liked to see a discussion on the difference between task specification during training a robot, and during testing a robot. These are quite different. In the "Get a bottle of water" example, saying that sentence at test time is perfectly sufficient. Any human would be able to interpret this, it is not as ambiguous as the authors are claiming. But during training, the task would need to be specified much more clearly than this, such as using goal images, demonstrations, or reward functions. There was no distinction made between the two, whereas I think that each has its own distinct issues that could be addressed in different ways.


In summary, the paper is an interesting read, and it did make me think about some of the open problems in machine learning. But I did not find myself thinking about robot learning itself in any depth. So other than having a reminder of all of these issues, the paper hasn't really made me think deeply about any new problems, and it hasn't really proposed any solutions to the problems it raises.

---

### Decision · Program_Chairs · 2021-10-01

**Decision:**

Accept

**Comment:**

The short paper brought up a large number of issues with an interesting position. It is very well written. The author is encouraged to further identify focal points and draw specific connection to robotics for discussion during the conference.